# Effects of Gamma Irradiation on the Fecundity, Fertility, and Longevity of the Invasive Stink Bug Pest *Bagrada hilaris* (Burmeister) (Hemiptera: Pentatomidae)

**DOI:** 10.3390/insects13090787

**Published:** 2022-08-30

**Authors:** Massimo Cristofaro, René F. H. Sforza, Gerardo Roselli, Alessandra Paolini, Alessia Cemmi, Sergio Musmeci, Gianfranco Anfora, Valerio Mazzoni, Michael Grodowitz

**Affiliations:** 1Biotechnology and Biological Control Agency (BBCA) Onlus, Via Angelo Signorelli 105, 00123 Rome, Italy; 2Italian National Agency for New Technologies, Energy and Sustainable Economic Development (ENEA), Via Anguillarese 301, 00123 Rome, Italy; 3European Biological Control Laboratory, United States Department of Agriculture, (EBCL USDA-ARS), 810 Avenue du Campus Agropolis, 34980 Montferrier-sur-Lez, France; 4Technology Transfer Center, Fondazione Edmund Mach, 38010 San Michele all’Adige, Italy; 5Research and Innovation Center, Fondazione Edmund Mach, 38010 San Michele all’Adige, Italy; 6Center of Agriculture, Food and Environment (C3A), University of Trento, 38010 San Michele all’Adige, Italy

**Keywords:** sterile insect technique, irradiation, sterility, biological control, insect pest, pentatomids

## Abstract

**Simple Summary:**

Controlling alien insect pests in cropping systems with no use of chemicals has always been challenging. Here, the first research studies to evaluate the use of irradiation to determine the feasibility of the sterile insect technique (SIT) approach to controlling the bagrada bug, *Bagrada hilaris*, are described. This work complements investigations on biological control for some of the major pentatomid pests, e.g., the brown marmorated stinkbug, the southern green stink bug, or the bagrada bug, using specific egg parasitoids. The complete sterility of males and females was reached with a minimum of 100 Gy gamma irradiation dose. This study documented how various irradiation doses impact the life history parameters of the bagrada bug, such as fertility, fecundity, and longevity. The results warrant further research to test the SIT directly on bagrada bug populations in the field or combine the SIT with a classical biological control program.

**Abstract:**

The bagrada bug, *Bagrada hilaris*, is an invasive insect pest in the family Brassicaceae that causes economically important damage to crops. It was originally present in Asia, the Middle East, and Africa, and was reported as invasive in the southwestern part of the US, in Chile, and on a few islands in the Mediterranean Basin. In its native range, *B. hilaris* is controlled by several egg parasitoid species that are under consideration as potential biological control agents. This research evaluated the impact of gamma irradiation on life history parameters, e.g., the fecundity, fertility, and longevity of *B. hilaris*, as a critical step towards assessing the feasibility of using the sterile insect technique against this recent invasive pest. Newly emerged adults of a laboratory colony originally collected from the island of Pantelleria (Italy) were gamma-irradiated. Life history parameters were evaluated at nine different doses, ranging from 16 Gy to 140 Gy. The minimal dose to approach full sterility was 100 Gy. Irradiation up to a maximum of 140 Gy apparently did not negatively impact the longevity of the adults. Even if both genders are sensitive to irradiation, the decline in fecundity for irradiated females could be exploited to release irradiated males safely to apply the SIT in combination with classical biological control. The data presented here allow us to consider, for the first time, the irradiation of bagrada adults as a suitable and feasible technique that could contribute to guaranteeing a safe approach to control this important pest species in agro-ecosystems. More research is warranted on the competitive fitness of irradiated males to better understand mating behavior as well as elucidate the possible mechanisms of sperm selection by polyandric *B. hilaris* females.

## 1. Introduction

The bagrada bug, *Bagrada hilaris* (Burmeister), is an invasive stink bug, native to Africa, Central Asia, and the Middle East [1,2,3]. It was recorded for the first time in the US in 2008 in California and since has expanded its distribution rapidly to southwestern US states, Hawaii, and Mexico [4,5,6,7]. Its presence has also been reported in Mediterranean Europe (Pantelleria Island, Italy, and Malta), Australia, Chile, and Southeast Asia [8,9]. It primarily feeds on Brassicaceae species, though it is known to have a relatively large host range, feeding and damaging agriculturally important food crops including corn (*Zea mays* L.; Poales: Poaceae), kale (*Brassica oleracea* L.; Brassicales: Brassicaceae), arugula (*Eruca vescicaria* (L.) Cav., Brassicales: Brassicaceae), sunflower (*Helianthus annuus* L.; Asterales: Asteraceae), among others [5,10], and caper plants (*Capparis spinosa* L.; Brassicales: Capparaceae) in southern Europe [11]. The pest’s impact is particularly severe when large numbers of aggregating nymphal stages induce excessive feeding damage by causing the destruction of apical meristems, young leaves, and terminal growing points [12]. There are few control strategies available to adequately suppress damaging populations of *B. hilaris* [13]. In the US, chemical control with pyrethroids and neo-nicotinoids demonstrated some impact, but the necessary multiple applications of these broad-spectrum insecticides have negative consequences for the economy and the environment [5,14]. Several egg parasitoid species, belonging to the genera *Trissolcus*, *Gryon*, and *Ooencyrtus*, are under consideration as potential agents in classical biological control programs against bagrada bugs [15,16,17]. These parasitoid species have been detected in Pakistan using sentinel eggs, an extremely valid biocontrol tactic based on exposing a consistent number of eggs of the target pest species for an amount of time suitable to obtain the egg parasitoid oviposition [17].

The sterile insect technique (SIT) is a species-specific pest control method based on the mass rearing, sterilization, and inundative releases of sterile insects (generally males) of the same pest species [18]. Irradiation, such as with gamma rays and X-rays, is used to sterilize mass-reared insects so that, while they remain sexually competitive, they cannot produce offspring [19,20,21]. Wild females that mate with sterile males do not produce offspring, thus preventing the growth of the pest population [18,19]. It is considered among the most environmentally friendly insect pest control method ever developed and there are no documented off-target effects reported [18,19].

However, the irradiation biology of Hemiptera remains to be fully studied. Currently, there are no field applications of SIT, mainly because the release of sterile phytophagous hemipteran adults of widespread highly invasive pest species could cause unwanted damage to host crop species [20]. Nevertheless, the opportune application of an SIT program is capable of achieving suppression and even eradication in particular geographic and/or infrastructural conditions (e.g., greenhouses, siloes, and isolated crop areas) if certain conditions related to the target pest biology (multivoltine, gregarious, suitable mating behavior, in the case of *B. hilaris*) and its response to the irradiation are met [21]. Moreover, for many hemipteran pest species, the use of irradiation to induce egg sterility can be applied in synergy with a classic biological control approach, based upon the use of co-evolved egg parasitoids. Recent data [22] has confirmed that sterile pentatomid eggs are a suitable substrate for the oviposition and full larval development of their egg parasitoids.

Interest in evaluating the feasibility of applying irradiation techniques in support of classic biological control to manage *B. hilaris* arose from the concept that classical biological control strategies for managing pentatomid pests are focused on the use of egg parasitoids [23,24,25,26]. Egg parasitoids, both in the area of origin of the target pest species or in the new environment, are detected and monitored by using newly oviposited sentinel eggs (lasting just 72 h in the field [25]) or frozen sentinel eggs (lasting 3–4 days [26]). The short exposition time is not the only limiting factor: manipulated eggs are less attractive and the distribution of live fertile eggs of pentatomid pest species in the new environment can be considered ethically incorrect and can unintentionally increase the dispersal of the target pest. This work should be considered the first step in the development of a new concept of “sterile sentinel eggs”, aimed at verifying if sterile eggs obtained by mating irradiated males with fertile females can be used in a classical biological control context to improve the management of this pest in a synergic manner [27,28].

This can be particularly true for *B. hilaris*, a pentatomid species that exhibits the unusual behavior of ovipositing in soil, which differs from all other pentatomid species [29,30]. For this reason, special emphasis has been placed on the scelionid egg parasitoid *Gryon aethirium* Talamas (formerly *G. gonikopalense* (Sharma)) [31], which is able to detect and oviposit on buried bagrada eggs [15,29].

Recent experiments evaluated the effect of irradiation on other invasive pentatomid pests, including the effects of irradiation on the biology of the newly emerged males of the brown marmorated stink bug (BMSB), *Halyomorpha halys* Stål [32], and the effects of high irradiation doses on the ultrastructure of the southern green stink bug (SGSB), *Nezara viridula* L. [33].

The central research questions raised in this study are how effective gamma irradiation is in inducing sterility in both sexes of bagrada and what effects increasing irradiation doses have on its longevity, fecundity (number of eggs oviposited by irradiated females), and fertility (number of hatched eggs in comparison with the number of eggs oviposited). This current research is the initial step of a wider program seeking to assess the feasibility of an SIT approach in synergy with classic biological control for a strategic plan for suppressing *B. hilaris* on the Italian island of Pantelleria.

## 2. Materials and Methods

### 2.1. Insects and Rearing 

Adults of *Bagrada hilaris* were field collected from cropped caper plants (*Capparis spinosa* L.) in Scauri (36.7722; 11.0606; 22 m a.s.l.) on Pantelleria Island, Italy during the summer of 2019 and 2020. Living material (nymphs and adults) for a total of ca. 100 individuals was transferred to quarantine facilities located at the USDA-ARS, EBCL, Montferrier-sur-Lez, France, for preliminary screening to develop correct rearing protocols. Insects were reared in cloth insect cages (BugDorm^®^, Taichung, Taiwan) (60 × 60 × 60 cm, 680 μm mesh opening) in a quarantine greenhouse (temperature ranged from ca. 25 °C (night) to ca. 35 °C (day); light was supplemented with sodium lamps (L/D 12:12; RH: 30–70%)) [15,34]. Various host plant species were used, e.g., broccoli, mustard, and false rocket, to create suitable environmental and biological conditions to rear multiple successive generations in the laboratory [15].

For the irradiation bioassays, about 1500 overwintering adults were collected at the same locality during the last week of October and the second week of November 2020 and subsequently reared at the quarantine facility of Fondazione Edmund Mach, San Michele all’Adige, Italy. The insects were transferred into BugDORM^®^ cloth cages (30 × 30 × 30 cm, 680 μm mesh opening) and fed *Brassica oleracea* L. var. *gemmifera* (Brussels sprouts). The climatic conditions were similar to the parameters used at the USDA-ARS EBCL quarantine facility, except that only artificial light was provided. Two open plastic Petri dishes (9 cm diameter) filled with fine sand were located at the bottom of the rearing cages and served as the oviposition site. Egg eclosion was determined using a light microscope (Olympus SZX-ILLB200; Olympus, Hamburg, Germany).

### 2.2. Irradiation

Fifth instar nymphs were singularly isolated in separate plastic 50 mL Corning^TM^ Falcon tubes covered with organdie mesh for transpiration to allow the collection of 50 virgin, newly emerged *B. hilaris* bugs for irradiation. Groups of ten male and ten female newly emerged individuals were confined in two Petri dishes of 9 cm in diameter. All groups were irradiated with 16, 24, 32, 50, 60, 80, 100, 120, or 140 Gy at the ^60^Co gamma facility at ENEA (Italian National Agency for New Technologies, Energy, and Sustainable Economic Development, Rome) [35]. The dose rate was 175.03 Gy/h (2.92 Gy/min). Insects in the control (0 Gy) treatment group were exposed to the same environmental conditions (temperature ranged from 25 to 35 °C; L:D = 12:12: RH: 30–70%) as the irradiated insects to exclude any effects on the survival rate of the treated bugs due to different rearing conditions.

### 2.3. Crosses and Assessments

A total of three treatments and a control were used. All crosses were carried out using virgin individuals. These include:(a)Irradiated male × Fertile female (IM/FF);(b)Fertile male × Irradiated female (FM/IF);(c)Irradiated male × Irradiated female (IM/IF);(d)Fertile male × Fertile female (FM/FF) (zero-dose control).

A minimum of five replicates were performed for each treatment. Crosses were accomplished by confining each pair in a 500 mL transparent glass jar covered with a 680 μm white polyester mesh, with a 5 cm-diameter open Petri dish filled with fine sand as an oviposition site. Each pair was fed with a single Brussels sprout. Every two days (or 3 days during weekends) the pairs were checked to record survival rate and the Petri dishes were inspected under a stereomicroscope (Olympus SZX-ILLB200; Hamburg, Germany) at 20× to record the numbers of eggs oviposited. The collected eggs were transferred to filter paper housed in a 3 cm-diameter Petri dish, kept under the same climatic conditions, to evaluate the eclosion rate. The oviposition experiment was terminated with the death of the adults or if the couples were still alive after three weeks. Living adults were kept after this period for another 3 weeks to evaluate longevity without collecting any additional data on oviposition and egg eclosion rate during this period (see Section 2.4.2).

### 2.4. Statistical Analysis

#### 2.4.1. Data Modeling on Reproductive Parameters 

A Generalized Linear Model (GLiM) [36,37] was performed according to an ANCOVA design. The number of hatched eggs, the number of laid eggs, and the rate of hatched eggs were the response variables, while the dose and the type of cross were the explanatory variables. The irradiation dose was treated as the continuous explanatory variable (dose Gy from 0 to 140), while the type of bug cross was treated as the categorical explanatory variable. The GliM model was chosen after the assessment of a significant deviation from normality by the Shapiro–Wilks test and after an analysis of fit, residuals, and a q–q plot. Since the data referring to oviposition and egg hatching show high overdispersion, a quasi-Poisson model with a log*_e_θμ_i_* link function was used, where *μ_i_* is the mean expected response and *θ* is the estimated overdispersion parameter. A quasi-binomial model with a “logit” link was fitted to the data on egg hatching (proportion), and these were analyzed as the proportion between the unhatched and hatched eggs. In the latter case, in order to stabilize variance, a log transformation was applied to the explanatory variable “dose” [37]. The models were fitted to assess whether (1) there was a relationship between the irradiation dose and the response variables and (2) if the different levels of the categorical predictor had different slopes, thus indicating a differential effect of the gamma rays on the different crosses. Only IM/FF, FM/IF, and IM/IF levels were considered in the analysis, since the FM/FF control group was treated as the zero dose of the 3 treatments, thus allowing the 3 groups to keep their own intercept. This model design allowed us to treat the dose effect as a covariate term among all dose treatments from 0 to 140 Gy and to avoid aliased terms. According to this model design, only the variable dose and its interaction with the categorical variables makes sense for the interpretation of the outcomes, while no significant main effects should be expected for the categorical variables. 

#### 2.4.2. Data Modeling for the Dose–Response Estimation in Egg Hatching

A curve model was applied with functions drm and mselect in the **drc** package (Dose–Response Curves, Version 3.0-1) [38], in order to define the Effective Dose values in the 3 types of crosses. The best model was chosen by comparing the log-likelihood values, Akaike’s Information Criteria (AIC), lack of fit, and residual variance of all models. The % of response of egg hatching was compared to the control and was considered as the response variable. This was expressed as Pt/Pc × 100 (Abbott correction [39]), where Pt is the proportion of hatched eggs for the treatment and Pc is the proportion of egg eclosion for the control. The dose treatment (treated as a continuous variable) and the type of crosses (treated as categorical variables) were the explanatory variables. The dataset was fitted to a non-linear Weibull model, chosen after the analysis of diagnostic parameters described above:*y* = *c* + (*d* − *c*){−exp{−exp[*b*(log(*x*) − log*e*)]}}. 

For this model, *d is* the upper limit, *c* is the lower limit, *b* is the slope, *e* is the inflection point, and *x* is the irradiation dose (Gy). The *c* parameter (the lower limit) was fixed to zero (W1.3 model), as it was assumed there were no eclosed eggs for the values of dose irradiation for lim f(*x*) → ∞. The *d* parameter was fixed to the same value for the three types of crosses, since no differences were assumed for the 3 types of crosses at dose 0. 

The distribution of errors was corrected by the Box–Cox transformation. Sterilizing effective doses ED50, ED90, ED95, and ED99 % were determined with their respective 95% confidence intervals and the estimated doses from the different crosses were compared using Student’s *t*-test (*p* < 0.05).

The statistical significance of the treatment (the type of cross), was verified by comparing the whole model without treatments with that of treatments using an ANOVA simple test by the anova.drc function, while the comparisons among groups for model parameters and ED values were performed by the compParm and Edcomp functions, respectively. In the case of the laid and hatched numbers of eggs, a Weibull model was also applied for plotting the curve trends. All graphs were constructed using the plot function in the graphics package [40] in the statistical environment R [41]. 

#### 2.4.3. Longevity of Adults 

A survival analysis was performed using SPSS inc. PASW Statistics 17.0, Release Version 17.0.2 (2009) [42] on data regarding adult longevity, and the survival curves were analyzed by the Log-rank test (Mantel-Cox). The test was applied separately for males and females in comparison to longevity as a function of dose. Due to the fact that we increased the number of replicates for the 4 higher doses, only 80, 100, 120, and 140 Gy doses were tested for the adult lifespan, the range when full sterility was recorded. The effects of gamma irradiation on longevity were evaluated by comparing the survival of the adults irradiated at these doses with those of the untreated adults. The experiment was terminated with the death of the adults or if they were still alive after six weeks. We assumed that if irradiated adults were still alive after this period, their longevity would be similar to the untreated controls.

## 3. Results

### 3.1. Fecundity and Fertility

#### 3.1.1. Effects of Gamma Irradiation on Oviposition and Egg Eclosion

Gamma irradiation strongly interfered with fecundity (number of eggs oviposited by irradiated females) and with fertility (number of hatched eggs in comparison with the number of eggs oviposited). The irradiation of females had a clear effect on their fecundity, as the ability of irradiated females to complete oviposition was reduced with increased irradiation dose, but fecundity remained high when only the males were irradiated (Figure 1b; Table A1 and Table A2 in Appendix A). Similarly, the viability of the sperm of irradiated males declined with increasing irradiation dose (Figure 1a), with a progressive reduction in the egg eclosion rate (Figure A1, Table A1, Table A2 and Table A3 in Appendix A).

#### 3.1.2. Data Modeling on Reproductive Parameters

Most of the data variability was explained by the predictor variables of the three GLiM models performed on oviposited eggs, eclosed eggs, and egg hatching rate data. In fact, the ratio between residual deviance and null deviance was 0.19 in the case of egg eclosion (fertility), 0.40 in the case of the oviposited eggs (fecundity), and 0.28 in the case of the egg hatching rate. Table 1 shows the overall results of the analysis of deviance, while Table 2 shows the estimated coefficients of the models. As regards coefficients, the IM/FF cross was set to zero and treated as the reference level for the remaining two crosses. Any significant main effect of the cross was detected for both the response variables, as expected from the model design (see statistical methods in the Section 2.4.1). As regards the analysis of deviance performed on fertility data (Table 1), irradiation dose showed a strong and highly significant negative effect on fertility (χ^2^ = 54.17, *p*-value = 1.8 × 10^−13^ ***), whereas only a weak tendency was detected for the interaction term. Nevertheless (Table 2), the IM/IF cross showed a weak but significant interaction with the dose effect (t = −2.062, *p*-value = 0.041 *, Table 2), suggesting a more rapid decline in fertility compared to the IM/FF cross. As regards fecundity data, the analysis of deviance (Table 1) showed a significant and strong interaction between dose and type of cross. In particular (Table 2), the t-value was strongly negative for both crosses compared to the IM/FF cross (−7.763, *p*-value = 6.80 × 10^−13^ *** for FM/IF, and t = −6.872, *p*-value = 1.08 × 10^−10^ *** for IM/IF), thus indicating that the dose effect was strictly dependent on the type of cross, with the suppression of fecundity only in the case of the IM/IF and FM/IF crosses (Figure 1b). With regards to the proportion of egg hatching, the analysis of deviance (Table 1) showed a very strong effect of dose and a slighter but significant effect of the interaction between dose and crosses. In fact, as Table 2 shows, a significantly lower effect of suppression in the egg eclosion rate was observed in the FM/IF cross than in the IM/FF cross.

#### 3.1.3. Data Modeling on the Percentage of the Response of Egg Eclosion

According to the AIC, the residual variance values, and the other parameters provided from the mselect function, the W1.3 curve was found to be the best model for the % of response of egg eclosion. Table 3 describes the W1.3 model, its coefficient parameters, and their statistical significance. Except for the *e* parameter of curve IM/IF, the remaining parameters were significant for the three crosses (Table 3), indicating that the model was robust to adequately explain the trends of curves. Figure 2 visualizes the W1.3 model and the differential effects recorded on the three bug crosses as a function of dose. The rate of egg eclosion was strongly affected by gamma irradiation but also influenced by the type of cross (F ANOVA = 3.49, *p* = 0.0098), therefore indicating that the three curves had different slopes (parameter *b*) and inflection points (parameter *e*) (Table 3). In particular, a comparison of the parameters in Table 3 displays that the ***e*** parameter was significantly lower in the IM/IF cross in comparison to the others (t-value = 3.48, *p*-value = 0.0006 for the IM/FF–IM/IF comparison and t-value = 2.50, *p*-value = 0.013 for the FM/IF- IM/IF comparison), indicating a different inflection point with a more rapid decrease in response. However, the lack of significance for the “e” parameter in the IM/IF cross suggests caution in this case, and further experimental evidence with a larger sampling is needed. Concerning the *b* parameter (the curve slope), this was lower than 1 in the case of the FM/IF and IM/IF crosses (Table 3), although the statistical significance was reached only in the case of the IM/FF–IM/IF comparison (t-value = 1.56 *p*-value = 0.120 for the IM/FF- FM/IF crosses and t-value = 2.55, *p*-value = 0.011 for the IM/FF- IM/IF crosses). A parameter *b* lower than 1 indicated that the rate of the decrease in egg eclosion, although rapid at low doses, slowed down as the dose increased (Figure 2). The Effective Dose values (ED) estimated for the three cross treatments can be found in Table 4. As regards the FM/IF and IM/IF crosses, it was possible to draw reliable ED values only up to ED90 doses (50.64 ± 11.38 and 115.83 ± 22.9, respectively) but not in the case of the ED95 and ED99; the latter was almost unreliable (ED99 296.07 ± 173.81 and 346.46 ± 159.26, for IM/IF and FM/IF, respectively). On the contrary, the data referring to the IM/FF cross reached full sterility at 100 Gy, thus allowing a more reliable estimate of effective dose up to ED99. In particular, they were 21.94 ± 5.24, 64.43 ± 7.14, 81.59 ± 11.05, and 120.00 ± 24.16 Gray for ED 50, ED90, ED95, and ED99, respectively.

In summary, the analysis of fecundity and fertility curves indicates that the IM/FF treatment had the strongest impact on the suppression of fertility when compared to the FM/IF and IM/IF treatments (see also Figure A1, Table A1, Table A2 and Table A3 in Appendix A for the three types of crosses). The response was more homogeneous and the irradiation did not interfere with mating or fecundity. In fact, the number of oviposited eggs, although highly variable, was comparable on average with the control at all the doses tested (Figure 1), while at 24 Gy, egg fertility decreased to about 50% of the control (52.5% ± 18.6 s.e.) and then approached zero beginning with 100 Gy and higher (Table A1, Figure 1). On the contrary, the number of eggs oviposited by irradiated females significantly decreased beginning at 32 Gy, i.e., at 32 Gy, 25.8 ± 8.4 s.e. eggs, as compared to the control at 141.6 ±11.5 s.e. eggs, while the sterility rate became more inconstant than with the IM/FF treatment (compare Table A2 and Table A3 for FM/IF and IM/IF vs. IM/FF).

The egg-eclosion rate clearly indicated that the sperm of males irradiated at 50 Gy reduced fertility to only 22.5% ± 7.1 s.e. as compared to the control; at 60 Gy, sterility approached 90%, and at 100 Gy and 120 Gy, almost all sperm of the irradiated males was sterile. 

### 3.2. Longevity of Adults

Giving priority to the evaluation of the bioassays on fecundity and fertility, the evaluation of adult longevity was focused only on the effects of the last four irradiation doses (80, 100, 120, and 140 Gy), where full sterility was recorded. The longevity values at dose 0 were 21.1 ± 1.9 (s.e.) days and 28.7 ± 2.2 days for males and females, respectively. The longevity values of the untreated were similar or even lower to those recorded from irradiated adults, as in the case of the irradiated males at 80 Gy (Table 5). As the Kaplan Meyer curves show, the highest longevity values were recorded for some untreated individuals, but most of those irradiated showed a trend very similar to the untreated ones or higher, at least for the first 30 days of the bioassay (Table 5 and Figure A2 in Appendix A).

## 4. Discussion

By gradually increasing the irradiation doses on males in the study, we observed a negative impact on reproduction, with almost complete female sterility at the highest dose. As such, male irradiation at 22 Gy induced 50% of estimated sterility on newly emerged bagrada bug adults while 90% sterility was achieved by irradiating newly emerged males with an estimated dose of 64 Gy (Table 4). In addition, we observed almost complete sterility with doses starting from 100 Gy upwards, resulting in 99% egg sterility at this dose. However, we observed occasional egg eclosion with the first instar nymphs unable to move and feed, resulting in death within three days. 

Obtaining sterile eggs from irradiated males and/or females provides a key element that can be used in synergy with a classic biological control approach, based upon the use of co-evolved egg parasitoids, as recently shown with the brown marmorated stinkbug [22]. This approach supports the new concept of developing a small-scale biofactory for the production of sterile sentinel eggs, to be used to detect new egg parasitoids and to monitor the dispersal of the species released to control *B. hilaris* in the new environment. Knowing the optimal irradiation doses for bagrada adults may help in designing future experimental SIT approaches. We acknowledge that this is a very premature stage of an SIT approach, but our study draws the line on how best to guarantee the 100% sterility of an irradiated cohort of bagrada bugs in case we envision mass rearing, irradiating, and releasing in open field conditions.

Interestingly, the effects of irradiation on reproduction were slightly different between males and females, not in terms of fertility but in terms of fecundity: the number of eggs oviposited by the irradiated females was negatively affected by the irradiation dose, declining from 60% at 50 Gy up to 80% at 100 Gy, whatever the male status (fertile or irradiated).

Conversely, the sperm of irradiated males did not have any significant effect on the female fecundity. The reproductive sterility is typically induced by exposure to X-rays, electron beams, or gamma rays from a ^60^Co or ^137^Cs source [43], which all cause chromosomal damage. Sterility is usually permanent, although the irradiated males of some species may, over time, regain at least partial fertility, especially following multiple matings [19,44]. Irradiation doses suitable for an SIT program that sterilizes males do not induce immobility and/or physiological deformations in the sperm [45]. However, in irradiated females, they can destroy oogonial cells, but the radiation sensitivity of oocytes varies with such factors as maturity and meiotic stage. As a result, the females of some species may retain a degree of fertility after irradiation, especially when treated late in ovarian development [20].

In contrast to females, the mating competitiveness of sterile males is a function of their mating propensity and mating compatibility and is a crucial component of the sterile insect quality. Irradiation could be responsible for reducing insect quality in some cases: thus, the ability of released sterile males to compete for mating is critical [20]. 

Different from what was observed in *H. halys*, where longevity declined significantly at irradiation doses higher than 24 Gy [21,29], irradiation even at the four highest doses used did not interfere with the longevity of *B. hilaris*. The survival rates of the irradiated adults exposed to 100, 120, or 140 Gy did not differ from the control, whereas male longevity was apparently greater at 80 Gy in comparison with the untreated individuals. If this is applied in a wider approach with releases of irradiated bagrada males, then we can consider that either a stable or increased longevity may favor the success of an SIT approach. In fact, sterile males will remain in the environment for a long period of time, mating with fertile wild-type females to deposit sterile eggs over the course of the vegetative growth of the target crop species to protect, e.g., cole crops, caper plants, or rooibos tea. In addition to stopping or reducing the regeneration of bagrada in the environment, the presence of sterile eggs may attract resident or intentionally released egg parasitoids. This double action of irradiation and biological control needs to be further studied with field trials.

There is not a “perfect” target pest eligible for the SIT [46]. The history of the SIT has been focused mainly on two orders: Diptera and Lepidoptera [47]. Other orders and families, such as Hemiptera and Pentatomidae, were disregarded from SIT applications. The fact that the feeding behavior of (sterile) adults induces important economic damage to their host plant crops always remained an issue [19]. However, this trend changed some years ago, through the preliminary screening of the effects of gamma irradiation on the longevity and fertility of *H. halys*, to consider the possibility of using SIT to control BMSB. Previous work on *H. halys* and another pentatomid species, *N. viridula*, had already demonstrated the potential of SIT in managing alien pentatomid pests under favorable ecological conditions [29,30]. The use of the SIT with *H. halys* in an area-wide IPM context was mainly considered for potential eradication or suppression programs in the case of early detection in New Zealand [21,29]. 

In the same way, the results presented in this work support the idea that *B. hilaris*, a pentatomid with a narrow climatic range, could be suppressed or even eradicated in specific geographic areas and ecological conditions. The invasive Italian population of *B. hilaris* confined on the small island of Pantelleria [11] presents the opportunity to use the SIT in the future as a key component of an area-wide program in synergism with other least-toxic strategies. 

In a traditional SIT program, the development of mass rearing facilities able to produce large numbers of sterile adult males that are sexually competitive with the wild-type is a prerequisite [46,48]. However, the mass-rearing process can also promote genetic drift, inducing genotypic differences between wild and laboratory populations [49]. In other words, captivity can cause dramatic shifts in strain genetics, including reductions in diversity (heterozygosity), with a clear increase in the lines better adapted to a given experimental condition [50,51]. In the case of bagrada, the hypothesis of using wild populations is envisioned. As such, massive collections of wild bagrada bugs, especially during the fall and winter when the bug is aggregating in huge numbers around the senescent host plants (MC and RFHS, unpublished), are highly realistic. Field observations carried out with the invasive Italian population during the winter of 2020–2021 confirmed the aggregation behavior of *B. hilaris*. Thus, the aggregation behavior and diapausing periods must be considered key periods for targeting bagrada control [47,52]. In some cases, these massive populations can be used as “small-scale mass-rearing natural biofactories” to provide and release wild-type sterile insects [47,49].

The results presented here warrant continued research to completely evaluate the performance of irradiated adults as we document how various irradiation doses impact the life history parameters of the bagrada bug. We began with low irradiation doses (16 Gy), increasing the dose according to physiological, biological, and behavioral responses, as previously described for BMSB [29,30]. A total of nine different doses (from 16 to 140 Gy) were evaluated on the newly emerged bagrada adults of both sexes. After BMSB [29], this is the second example of measuring the impact of irradiation on stinkbugs. 

Finally, mating in *B. hilaris* is subordinated by female acceptance and mediated by the “correct” male courtship behavior [53] and the production of a pheromone [54]. Choice bioassays will be necessary to understand if irradiated males can compete with fertile males in terms of mating. Since the female has polyandric behavior [55], additional screening must be carried out to evaluate the presence of cryptic sperm competition. In addition, feeding bioassays must be performed to provide a correct evaluation of the impact of the irradiated males, when released into the environment.

## 5. Conclusions

For the first time, the data presented here allow the consideration of the gamma irradiation of bagrada males and females as a suitable and feasible technique. The effects are optimal on sterile egg production when the appropriate dose of 100 Gg and above is applied to adults, with no emergence and survival of bagrada nymphs. If applied in the future, this will guarantee a safe approach without adding new pests to the agro-ecosystems to be controlled. The absence of a negative impact on the longevity of irradiated adults brings new insights into both the reduction in the bagrada population in the long term and the possibility for sterile eggs to be parasitized by wild egg parasitoids (RFHS, unpublished data; GR, personal communication). Field stinkbug egg exposures are an efficient technique to catch new egg parasitoids and have recently provided preliminary promising data with *B. hilaris* in invaded areas [15,56]. Using sterile sentinel eggs would then be a safe strategy for collecting parasitoids both in the native and introduced range, without bringing additional bagrada nymphs emerging from sentinel eggs into the environment. This approach is still prospective but may echo preliminary studies with *H. halys* [21]. 

In conclusion, the potential for a future SIT program as a component in an area-wide program to manage *B. hilaris* and other pentatomid pest species exists. Further research is needed to ascertain the geographical, agronomic, and socio-economic conditions for specific SIT field applications. Integrating classical biological control would enhance the impact of pentatomid area-wide programs involving SIT.

## Figures and Tables

**Figure 1 insects-13-00787-f001:**
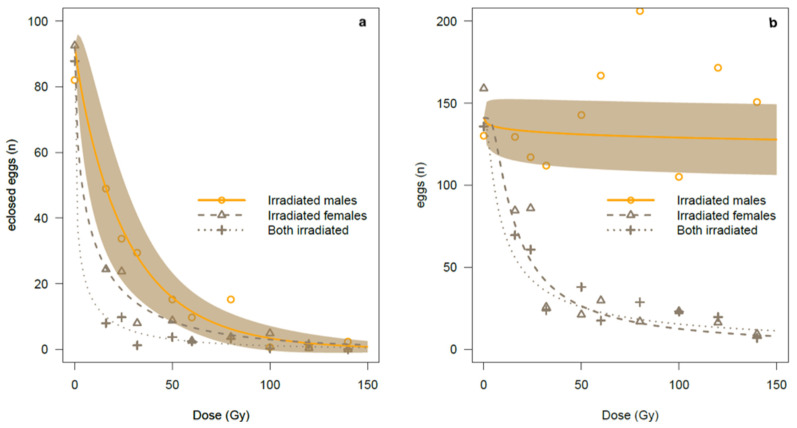
Comparison between three treatments (irradiated males only, irradiated females only, and both genders irradiated) on the number of eggs eclosed. Data were fitted by a Weibull model. For clarity, the 95% CI is shown only for the curve *irradiated males* × *fertile females*.

**Figure 2 insects-13-00787-f002:**
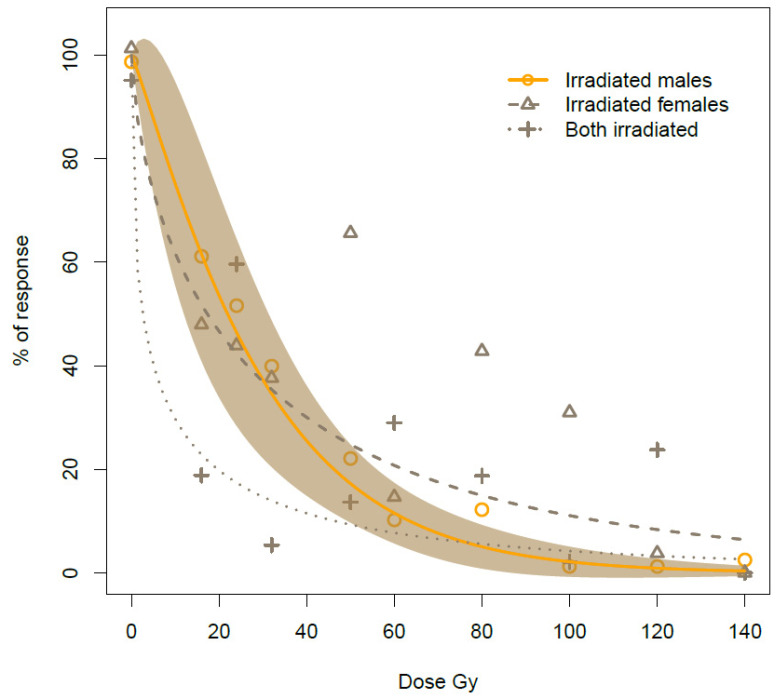
Weibull-1 curves (*c* parameter set to 0) calculated for the percentage of the response of egg eclosion for the three tested crosses (*irradiated males* × *fertile females; irradiated females* × *fertile males; both irradiated*). The 95% CI is shown only in the case of the irradiated males x fertile females curve, for clarity.

**Table 1 insects-13-00787-t001:** Analysis of deviance of the GLiM models. (Type III). The overall effect of dose and cross types on the number of eclosed eggs, the number of oviposited eggs, and the proportion of eggs hatched per female are reported.

Response Variable	Model Terms	χ^2^	df	*p*-Value
	dose	54.17	1	1.8 × 10^−13^ ***
Eggs eclosed	cross	0.214	2	0.8972
	dose × cross	5.707	2	0.0576
	dose	1.045	1	0.3066
Eggs oviposited	cross	1.509	2	0.4703
	dose × cross	110.36	2	<2.2 × 10^−16^ ***
	dose	263.80	1	<2.2 × 10^−16^ ***
Proportion of hatched eggs	cross	3.74	2	0.1544
	dose × cross	12.64	2	1.8 × 10^−3^ **

Levels of significance are reported in the table according to the conventional notation by asterisks: no symbols, *p* > 0.05; *p* ≤ 0.01 **; *p* ≤ 0.001 ***.

**Table 2 insects-13-00787-t002:** Coefficient estimates of the GLiM models for the number of eclosed eggs, the number of oviposited eggs, and the proportion of eggs hatched per female. The IM/FF cross does not appear in the table since this parameter is set to zero and matters as a reference level; thus, the FM/IF and IM/IF estimates are compared with this group.

Response Variable	Levels	Estimate	s.e.	t-Value	*p*-Value
	intercept	4.465	0.170	26.33	<2.2 × 10^−16^ ***
Eggs eclosed	dose	−0.033	0.006	−5.32	3.4 × 10^−7^ ***
	FM/IF	0.024	0.243	0.10	0.922
	IM/IF	−0.090	0.256	−0.35	0.725
	Dose × FM/IF	−0.012	0.011	−1.06	0.289
	Dose × IM/IF	−0.036	0.017	−2.06	0.041 *
	intercept	4.855	0.094	51.81	<2.2 × 10^−16^ ***
Eggs oviposited	dose	−0.001	0.001	1.02	0.307
	FM/IF	0.0237	0.243	0.09	0.573
	IM/IF	−0.0901	0.256	−0.35	0.455
	Dose × FM/IF	−0.0117	0.011	−1.06	6.8 × 10^−13^ ***
	Dose × IM/IF	−0.0357	0.017	−2.06	1.1 × 10^−10^ ***
	intercept	−0.754	0.170	−4.44	1.7 × 10^−5^ ***
Proportion of hatched eggs	dose	0.7407	0.056	13.31	<2.2 × 10^−16^ ***
	FM/IF	0.349	0.226	1.540	0.125
	IM/IF	0.369	0.233	1.583	0.115
	Dose × FM/IF	−0.306	0.088	−3.48	6.4 × 10^−4^ ***
	Dose × IM/IF	−0.017	0.110	−0.16	0.875

Levels of significance are reported in the table according to the conventional notation by asterisks: no symbols, *p* > 0.05; *p* ≤ 0.05 *; *p* ≤ 0.001 ***.

**Table 3 insects-13-00787-t003:** Model fitted for % of response: Weibull (type 1) with a lower limit at 0 and with the intercept fixed to 100% (crossings of *Bagrada hilaris*: IM/FF: irradiated virgin male × fertile virgin female; FM/IF: fertile virgin male × irradiated virgin female; IM/IF: irradiated virgin male × irradiated virgin female). The parameter *b* describes the slope, while the parameter *e* is the inflection point of the curve. Parameters followed by different letters are significantly different for *p* = 0.05 according to the t-statistics.

	Estimate		±s.e.	t-Value	*p*-Value
*b*:IM/FF	1.17043	a	0.26005	4.5007	1.288 × 10^–5^ ***
*b*:FM/IF	0.73009	ab	0.20332	3.5908	0.0004366 ***
*b*:IM/IF	0.48913	b	0.13945	3.5075	0.0005855 ***
*e*:IM/FF	29.91119	a	6.0447	4.9483	1.861 × 10^−6^ ***
*e*:FM/IF	27.92878	a	8.83973	3.1595	0.0018863 **
*e*:IM/IF	6.89095	b	4.48994	1.5348	0.1267950

Levels of significance are reported in the table according to the conventional notation by asterisks: no symbols, *p* > 0.05; *p* ≤ 0.01 **; *p* ≤ 0.001 ***.

**Table 4 insects-13-00787-t004:** Estimated effective dose (ED) calculated for the percentage of response in *Bagrada hilaris* eggs hatching in the three groups of crosses. IM/FF: irradiated male × fertile female; FM/IF: fertile male × irradiated female; IM/IF: irradiated male × irradiated female. Standard error (s.e.) and error limits are reported at the 95% percentile.

	ED (%)	Estimate (Gy)	±s.e.	Lower	Upper
IM/FF:	50	21.94	5.24	13.69	35.18
IM/FF:	90	64.43	7.14	51.77	80.18
IM/FF:	95	81.59	11.05	62.45	106.60
IM/FF:	99	120.00	24.16	80.63	178.59
FM/IF:	50	17.36	7.50	7.395	40.76
FM/IF:	90	115.83	22.90	78.39	171.13
FM/IF:	95	175.58	50.36	99.65	309.34
FM/IF:	99	346.46	159.26	139.78	858.76
IM/IF:	50	2.38	2.48	0.30	18.71
IM/IF:	90	50.64	11.38	32.49	78.93
IM/IF:	95	99.01	28.87	55.67	176.09
IM/IF:	99	296.07	173.81	92.88	943.76

**Table 5 insects-13-00787-t005:** Longevity (mean ± s.e.) of irradiated *Bagrada hilaris* males and females at high doses (80 Gy upwards) compared with the untreated control. Analysis of Log-rank (Mantel–Cox) was performed for mean separation. Means followed by different letters are significantly different between them.

	Longevity (days)							
Dose (Gy)	Male	±s.e.		n	Female	±s.e.		*n*
0	21.1	1.9	b	19	28.7	2.2	ab	38
80	32.5	1.4	a	6	40.4	2.4	a	8
100	23.8	1.7	b	14	26.7	3.0	ab	21
120	26.0	3.2	b	7	27.3	5.5	ab	10
140	16.5	2.5	b	8	21.0	2.4	b	10

## Data Availability

The data presented in this study are available on request from the corresponding author.

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
