# Peer review of "Effects of Gamma Irradiation on the Fecundity, Fertility, and Longevity of the Invasive Stink Bug Pest Bagrada hilaris (Burmeister) (Hemiptera: Pentatomidae)"

_insects, 2022, doi:10.3390/insects13090787_

Round 1

Reviewer 1 Report (Previous Reviewer 1)

Thank you to the authors for their review of manuscript insects-1834456.  The description and use of statistical analyses were improved.  On the other hand, I still feel confused (and think many readers will be) on the concept of the SIT that is presented in the manuscript (I do not see how the authors addressed the first comment of my previous review).  My comments and suggestions follow.

L 22, suggest deleting ", such as stinkbugs,".

L 23, suggest "Here" instead of "Herein".

L 37-38, suggest, "... on fecundity, fertility and longevity of Bagrada hilaris, as a critical step towards assessing the suitability of using the sterile insect technique against this invasive insect pest" instead of "to determine the suitability of using the sterile insect technique (SIT) approach to control B. hilaris".  And then delete "Life history parameters such as fertility, fecundity and longevity were evaluated" on line 40.

At the end of the abstract indicate the conclusions of the work.  After all the work that was done, what do the authors conclude?  How does the study contribute to the bigger picture?

L 55, remove comma after "feeding".

L 55-56, include scientific names and botanical family of corn, kale, arugula, sunflower, and caper plants.

L 66, "have been detected" where?  In the natural geographical range of the bug or in invaded areas?

L 69, "The sterile insect technique (SIT)" instead of "The sterile insect technique, or SIT for short".

L 69-73, I think this paragraph lacks precision and clarity.  You start by saying that the SIT "is species-specific and there are no documented off-target effects reported".  But first, you need to indicate what is the SIT, and mention against which insect species the technique is applied, to understand the value of your statement.  Otherwise, the text makes no sense.  It is also mentioned that radiation is used to "sterilize mass-reared insects so that, while they remain sexually competitive, they cannot produce offspring".  However, note that the SIT is focused on the action of sterile males on wild females because a wild female that copulates with a sterile male leaves no offspring.

L 74-77, it seems to me that this paragraph contradicts the argument of the manuscript on the use of the SIT against Bagrada bug.

L 78-86, I was confused by the way this information is presented. As I mentioned in my previous review, the SIT involves more than just sterilizing insects. The SIT is based on the mass rearing and sterilization of the target insect, to release them in the field so that the sterile individuals mate with those of the wild population because the females that mate with a sterile male do not leave offspring and this prevents its reproduction and the growth of its population. Saying that irradiating an insect is the application of the SIT is not correct.  It seems to me that the authors are stretching the argument on the use of the SIT in combination with biological control because the SIT was not tested/used in this work.  As mentioned previously, authors need to properly flush out the significance of their work.

I was going to suggest authors cite the seminal paper of Knipling on the theoretical basis of the SIT, but the paper is already cited.  However, the article is incorrectly cited on line 68 which mentions biological control and sentinel eggs. It is important that authors carefully review the literature and make correct citations.

In their response to reviewer letter, the authors replied "I DO NOT AGREE" to my comment to move to the acknowledgments section the last paragraph of the introduction in which they mention that "The work was accomplished in Italy, in close cooperation with the USDA-ARS European Biological Control Laboratory (EBCL), Montferrier-sur-Lez, France".  However, they did not explain why they do not agree with my suggestion.  So, I cannot understand the rationale of the authors for including this sentence in the introduction.  As a reminder to authors, and as stated in the instructions for authors of Insects, "The introduction should briefly place the study in a broad context and highlight why it is important. It should define the purpose of the work and its significance, including specific hypotheses being tested. The current state of the research field should be reviewed carefully and key publications cited. Please highlight controversial and diverging hypotheses when necessary. Finally, briefly mention the main aim of the work and highlight the main conclusions. Keep the introduction comprehensible to scientists working outside the topic of the paper".  Based on this instruction, I still think that the last sentence of the introduction has no place in it.  But if the authors do not agree with my suggestion, they should explain why instead of just saying they do not agree.

Speaking of the introduction, there is no clear research question, hypotheses and predictions that allow a better understanding of the idea that the authors put to the test. This is important for readers to better understand and appreciate the relevance of the study.

L 166, "The number of" not "The n. of".

L 173, include the link function of the quasi-poisson GLM.

L 235, 236 and Figure 1, instead of saying graph on the right and graph on the left, include (a) and (b) in the figure panels and cite accordingly.

In the discussion, instead of beginning by repeating what you did, start by drawing readers’ attention to the most significant and relevant findings of your research. Unlike the introduction which moves from the general to the particular, the discussion should move from the particular to the general.  The discussion is the place to answer research questions and indicate if hypothesis and predictions were met.  But because the introduction lacks a clear research question, hypothesis and predictions, it is difficult to follow the line of reasoning for the study.  The discussion should also state limitations of the study and indicate how to address them in future studies.  Some of the information is somehow already present in the manuscript, but I feel that the way it is presented is hard to follow.

In the conclusions, what, exactly, is the "new concept of SIT for pentatomid insect pests"?  I recommend authors to close with stronger statements based on research results to clearly and succinctly transmit how the study bridged the gap of the specific topic of research.  That is, indicate clearly what knowledge does the study provide that was not known before and announce future research on the topic you will perform.

Author Response

Thank you to the authors for their review of manuscript insects-1834456.  The description and use of statistical analyses were improved.  On the other hand, I still feel confused (and think many readers will be) on the concept of the SIT that is presented in the manuscript (I do not see how the authors addressed the first comment of my previous review).  My comments and suggestions follow.

Dear Reviewer,

Thanks again for the detailed and valuable comments that really helped to improve the quality of our manuscript. We accepted all the remarks. In addition, we decided to change the title, removing the concept “SIT”.  We clearly understand your point of view and we agree that, even if the data we are presenting are very encouraging, it is too early to consider the screening for the sterile insect technique done. Moreover, following your suggestion, we improved the concept of Sterile Insect Technique in the introduction, and solve the mistake in the bibliography. 

Here below you can find a copy of your detailed list of comments, followed by our reply in italics.

L 22, suggest deleting ", such as stinkbugs,".  DONE

L 23, suggest "Here" instead of "Herein". DONE

L 37-38, suggest, "... on fecundity, fertility and longevity of Bagrada hilaris, as a critical step towards assessing the suitability of using the sterile insect technique against this invasive insect pest" instead of "to determine the suitability of using the sterile insect technique (SIT) approach to control B. hilaris".  And then delete "Life history parameters such as fertility, fecundity and longevity were evaluated" on line 40. DONE

At the end of the abstract indicate the conclusions of the work.  After all the work that was done, what do the authors conclude?  How does the study contribute to the bigger picture?  DONE

L 55, remove comma after "feeding".  ?  DONE

L 55-56, include scientific names and botanical family of corn, kale, arugula, sunflower, and caper plants. DONE

L 66, "have been detected" where?  In the natural geographical range of the bug or in invaded areas? DONE

L 69, "The sterile insect technique (SIT)" instead of "The sterile insect technique, or SIT for short".  DONE

L 69-73, I think this paragraph lacks precision and clarity.  You start by saying that the SIT "is species-specific and there are no documented off-target effects reported".  But first, you need to indicate what is the SIT, and mention against which insect species the technique is applied, to understand the value of your statement.  Otherwise, the text makes no sense.  It is also mentioned that radiation is used to "sterilize mass-reared insects so that, while they remain sexually competitive, they cannot produce offspring".  However, note that the SIT is focused on the action of sterile males on wild females because a wild female that copulates with a sterile male leaves no offspring.  DONE, fully agree: we forgot to describe the SIT concept 

L 74-77, it seems to me that this paragraph contradicts the argument of the manuscript on the use of the SIT against Bagrada bug. DONE.  We add a short sentence to highlight the concept that -as mentioned later at the end of the Introduction Chapter- the final goal will be to use -eventually- SIT in the field to suppress Bagrada in a very isolated area (Pantelleria Island), with very peculiar geographic conditions.

L 78-86, I was confused by the way this information is presented. As I mentioned in my previous review, the SIT involves more than just sterilizing insects. The SIT is based on the mass rearing and sterilization of the target insect, to release them in the field so that the sterile individuals mate with those of the wild population because the females that mate with a sterile male do not leave offspring and this prevents its reproduction and the growth of its population. Saying that irradiating an insect is the application of the SIT is not correct.  It seems to me that the authors are stretching the argument on the use of the SIT in combination with biological control because the SIT was not tested/used in this work.  As mentioned previously, authors need to properly flush out the significance of their work. DONE: fully agree, we changed the text (see lines 74-80)

I was going to suggest authors cite the seminal paper of Knipling on the theoretical basis of the SIT, but the paper is already cited.  However, the article is incorrectly cited on line 68 which mentions biological control and sentinel eggs. It is important that authors carefully review the literature and make correct citations.  DONE (sorry for the mistake)

In their response to reviewer letter, the authors replied "I DO NOT AGREE" to my comment to move to the acknowledgments section the last paragraph of the introduction in which they mention that "The work was accomplished in Italy, in close cooperation with the USDA-ARS European Biological Control Laboratory (EBCL), Montferrier-sur-Lez, France".  However, they did not explain why they do not agree with my suggestion.  So, I cannot understand the rationale of the authors for including this sentence in the introduction.  As a reminder to authors, and as stated in the instructions for authors of Insects, "The introduction should briefly place the study in a broad context and highlight why it is important. It should define the purpose of the work and its significance, including specific hypotheses being tested. The current state of the research field should be reviewed carefully and key publications cited. Please highlight controversial and diverging hypotheses when necessary. Finally, briefly mention the main aim of the work and highlight the main conclusions. Keep the introduction comprehensible to scientists working outside the topic of the paper".  Based on this instruction, I still think that the last sentence of the introduction has no place in it.  But if the authors do not agree with my suggestion, they should explain why instead of just saying they do not agree.  DONE (sorry): we moved the sentence in the acknowledgements

Speaking of the introduction, there is no clear research question, hypotheses and predictions that allow a better understanding of the idea that the authors put to the test. This is important for readers to better understand and appreciate the relevance of the study.  DONE: we improved the text (see lines 109-115)

L 166, "The number of" not "The n. of".  DONE

L 173, include the link function of the quasi-poisson GLM.  DONE (see lines 180-181) 

L 235, 236 and Figure 1, instead of saying graph on the right and graph on the left, include (a) and (b) in the figure panels and cite accordingly.  DONE

In the discussion, instead of beginning by repeating what you did, start by drawing readers’ attention to the most significant and relevant findings of your research. Unlike the introduction which moves from the general to the particular, the discussion should move from the particular to the general.  The discussion is the place to answer research questions and indicate if hypothesis and predictions were met.  But because the introduction lacks a clear research question, hypothesis and predictions, it is difficult to follow the line of reasoning for the study.  The discussion should also state limitations of the study and indicate how to address them in future studies.  Some of the information is somehow already present in the manuscript, but I feel that the way it is presented is hard to follow. DONE (see lines 415-423 and 443-448)

In the conclusions, what, exactly, is the "new concept of SIT for pentatomid insect pests"?  I recommend authors to close with stronger statements based on research results to clearly and succinctly transmit how the study bridged the gap of the specific topic of research.  That is, indicate clearly what knowledge does the study provide that was not known before and announce future research on the topic you will perform. DONE:  see lines 457-470)

Reviewer 2 Report (Previous Reviewer 2)

The authors have made the requested corrections and the manuscript is now acceptable for publication.

Author Response

According to what I know, I already replied in my previous revision to all the comments of the second reviewer

Reviewer 3 Report (New Reviewer)

The manuscript has been largely improved after the revision and I do not have further suggestions. The major suggestion from last time was the statistics. The statistic should be good enough in this version. However, for proportional data in this research, I bet the number of eggs used for hatching experiments might be different in different treatments (I did not find information from the manuscript). For example, you used 1000 eggs for one treatment and got 100 hatched, but in another experiment, you only have 10 eggs and got 1 hatched, even though the percentage of egg hatched were both 10%, the confidence level with different sample size is very different. So, I think using a Generalized linear model with binomial or quasibinomial distribution could be a better option since you can include the sample size information in the model.

Author Response

August 17, 2022

3rd Reviewer’s Comments and Suggestions for Authors

The manuscript has been largely improved after the revision and I do not have further suggestions. The major suggestion from last time was the statistics. The statistic should be good enough in this version. However, for proportional data in this research, I bet the number of eggs used for hatching experiments might be different in different treatments (I did not find information from the manuscript). For example, you used 1000 eggs for one treatment and got 100 hatched, but in another experiment, you only have 10 eggs and got 1 hatched, even though the percentage of egg hatched were both 10%, the confidence level with different sample size is very different. So, I think using a Generalized linear model with binomial or quasibinomial distribution could be a better option since you can include the sample size information in the model.

Reply to the 3rd Reviewer (August 17th, 2022):

Dear Reviewer,

We agree with you about your suggestion on statistics. To be more precise, we previously tried to weight data on the basis of the number of eggs, but we had some doubts due to the hatching rate remarkably high in the case of crosses FM/IF and IM/IF, when the number of laid eggs was lower (as if the number of hatched eggs was somehow independent from the number of laid eggs). So, we had some concerns that this information could be lost using weighted data. For this reason, we preferred to leave the percentages without weighting. However (we fully agree with you), a GLM analysis on proportional data should provide a better solution to this problem. For this reason, we added a GiLM model in the tables 1 and 2, assuming proportional data belonging to a quasibinomial family (due to overdispersion). In any case, we are still preferring to maintain the Weibull curve analysis, since it provides a more detailed information about curve shape. Instead, we omitted the inference with exact significance in the comparisons among Weibull curves (we apologize for do not have done this previously), because the e parameter was not significant in the case of IM/IF cross. Apparently, there are some discrepancies between GiLM and Weibull models (only the curve FM/IF statistically differs from the IM/MF cross in the case of GiLM model), probably due to the fact that the 2 models look at different aspects: the GiLM provides information about curve slope (assuming linearity), while Weibull model looks mainly at the curve shape. In conclusion, also to avoid confusion in the readers, we decided to report only more general considerations on curve shapes and relative parameters in the Weibull curve analysis, referring to the table 3 for further details (see also the changes in the lines 328-339).

Round 2

Reviewer 1 Report (Previous Reviewer 1)

It seems to me that the revised version of this manuscript goes in the right direction. Authors considerably changed the tone on the use of the SIT, but I feel that there are still some instances in the manuscript requiring attention of the authors to improve the quality of the manuscript.

Additional comments for the consideration of the authors follow:

General comments:

In the introduction it is explained about the risks of releasing sterile phytophagous Hemiptera because of the damage they can cause to host plants.  With this in mind, authors should make clear why they believe that in the case of the bagrada bug the benefits of releasing sterile insects outweigh the risks.  This is important to assess the value of the new text in lines 111-114, because in its current version it is not clear why authors think that releasing sterile bugs that damage host plants could synergize with classical biological control using egg parasitoids

The flow of ideas in the introduction could be improved.  For example, the first paragraph ends by talking about the use of parasitoids for biological control of the bagrada bug.  Then, the SIT is introduced and the irradiation of Hemiptera is mentioned in the second and third paragraph, respectively.  Then, in the fourth paragraph you go back to the classical biological control using parasitoids.  And in the fifth paragraph you return to the SIT... and so on.  Authors should make an effort to better link pieces of information.

Please strength and clarify the link between the use of sentinel eggs and classical biological control.

As previously suggested, include caveats of the study in the discussion.

Specific comments:

L 58-60, include botanical family of corn (Zea mays L.), kale (Brassica oleracea L.), arugula (Eruca vescicaria (L) Cav.), sunflower (Helianthus annuus L.) and caper plants (Capparis spinosa L.).

L 73-75, The sterile insect technique (SIT) is a species-specific pest control method based on the mass rearing, sterilization and inundative releases of sterile insects (generally males) of the same pest species [18]. 

L 77, After "... they cannot produce offspring [19–21].", include "Wild females that mate with sterile males do not produce offspring, thus preventing growth of the pest population (cite appropriate reference)".

L 84-87, Please explain how sentinel eggs can support classical biological control.  Do you plan to use sentinel eggs to collect native parasitoids that could attack the bug, then colonize them to establish a mass-rearing, and finally release them together with parasitoid species from the native range of the pest?

L 92, "is able to".

L 108-109, what, exactly, the authors refer to with "effective Gamma irradiation".  Please be specific.

L 111-114, before this sentence, it should be made clear how releasing sterile insects that can damage hots plants can synergize with classical biological control.  Also, how does the sentinel egg topic fits in here?

Author Response

 Dear Reviewer,

We agree with you about all your suggestions and comments. For this reason, we made several changes (highlighted in yellow in the new manuscript version). To be more precise, we added 4 more bibliographic references, we included additional sentences and, last but not least, we re-organized the text accordingly with your suggestion. 

In particular, we realized that you were absolutely correct reporting “the flow of ideas in the introduction could be improved”. Personally, I apologize for the mistake, and we really hope the work we have done to improve the organization of the text will meet with your approval.

In the uploaded file you can find our reply to each of all of your comments/suggestions.

Thanks

Massimo Cristofaro

Round 3

Reviewer 1 Report (Previous Reviewer 1)

In general, the manuscript continues to improve. But there are still details for the attention of the authors. These are some that I detected:

L 92-93, does this means that the SIT can be effectively applied against Hemiptera phytophagous pests when the target species is multivoltine, gregarious and (define "suitable mating behavior")?  What do you mean with "... response to the irradiation are met"?  That the adults can be sterilized?  Please modify text to improve clarity.

L 103, What happens to the eggs after 72 h (by the way, the abbreviation of hour is "h" not "hrs") or 3-4 days in the field?  Do they die or what happen to them?  Please modify text to improve clarity.

L 107-108, it would be clearer if you explain/define the new concept of sterile sentinel egg you put forth.

L 108-109, "verifying if mating sterile males with fertile females can be used in combination with classical biological control".  But you didn't really evaluate the link with biological control, did you?  Please modify text to improve clarity.

L 111-116, I think this paragraph can be safely removed because in its current form does not provide crucial information that allow to follow the line of reasoning.  Then, improve the link with the paragraph beginning with "This can be particularly true for B. hilaris, a pentatomid species that exhibits the...", because in its current form it is not clear what does "This" refers to.

L 195-197, A quasi-binomial model with __ link function was fitted to the data on egg hatch (proportion).

L 197-198, indicate why the log transformation was applied to the explanatory variable.

L 198, Models are "fitted", instead of "applied".

L 287, "the proportion of egg hatch".

In line 329 and elsewhere in the results, include p values/statistics after the word "significant".

L 467-473, Are there SIT programs for pentatomids?  Are you referring generally to the SIT and mass rearing or specifically to cases applied to pentatomids?   Please modify text to improve clarity.

Author Response

August 25th, 2022

Dear Reviewer,

Thanks a lot for your outstanding work in reviewing our manuscript. This time, we accepted (only) some of the comments/suggestions, replying in details to you telling why sometimes we decided to keep the text without your suggestions. Most of the times the reason of not acceptance was due to the fact that the concept you was suggesting was already present in the manuscript, eventually in a different page/chapter. Once again, I want to tell that most of your inputs have been very important to improve the quality of the paper. In other words, re-evaluating this long process of reviewing, I realized that this type of “special relationship” improved my and the other co-author skills and it has been very edifying and instructive. 

Best regards,

Massimo Cristofaro

In general, the manuscript continues to improve. But there are still details for the attention of the authors. These are some that I detected:

L 92-93, does this means that the SIT can be effectively applied against Hemiptera phytophagous pests when the target species is multivoltine, gregarious and (define "suitable mating behavior")?  What do you mean with "... response to the irradiation are met"?  That the adults can be sterilized?  Please modify text to improve clarity.   We agree that the sentence in the previous form is a little bit cryptic, and for this reason we add “in the case of B. hilaris” (Line 93). But some of the answers to the questions you are asking are still not published yet, and we believe that it will be correct to wait until they will be in press. There will be at least 2 more papers, one on “the differences between fertile males and irradiated males in mating courtships and feeding physiological patterns” and a second one on the study to verify the presence of a “last mating male sperm precedence”, that will be submitted in the next 3-4 months. Vice versa, the answers regarding the importance that bagrada bug is multivoltine and -similarly to BMSB- has a gregarious behavior for a suitable application of SIT, have been presented during the discussion (see Lines 484-487, where another bibliographic reference has been included).

L 103, What happens to the eggs after 72 h (by the way, the abbreviation of hour is "h" not "hrs") or 3-4 days in the field?  Do they die or what happen to them?  Please modify text to improve clarity.  Also for this comment, of course we fully agree to change “hrs” in “h” (thanks). Regarding the other questions/comments, fresh sentinel eggs can last only 72 h, because -according to the outdoor climatic conditions and to the Pentatomid species- there are big chances that after 4-5 days the eggs will hatch and the newly nymphs will start to search for suitable plants. This aspect has been already presented in the paper (see Lines 106-108) and for this reason we do not see the reason to add additional information regarding an aspect very important, but still in progress (there will be soon the submission of a paper on this topic, for BMSB and its parasitoids).

L 107-108, it would be clearer if you explain/define the new concept of sterile sentinel egg you put forth. (DONE: see the comment below).

L 108-109, "verifying if mating sterile males with fertile females can be used in combination with classical biological control".  But you didn't really evaluate the link with biological control, did you?  Please modify text to improve clarity. DONE: We changed the text to emphasize that the work we are presenting can be also supportive to classic biological control (see changes in the lines 109-111).   

L 111-116, I think this paragraph can be safely removed because in its current form does not provide crucial information that allow to follow the line of reasoning.  Then, improve the link with the paragraph beginning with "This can be particularly true for B. hilaris, a pentatomid species that exhibits the...", because in its current form it is not clear what does "This" refers to.  DONE: Actually, the first sentence has been removed and we transferred the second sentence below (new position: Lines 124-128). Of course, the order of the references has been changed accordingly.

L 195-197, A quasi-binomial model with __ link function was fitted to the data on egg hatch (proportion). DONE: See the changes at the Lines 204-205

L 197-198, indicate why the log transformation was applied to the explanatory variable. DONE: see changes at the line 206.

L 198, Models are "fitted", instead of "applied". DONE: Line 207

L 287, "the proportion of egg hatch". DONE: see Line 296 (and also lines 316 and 321)

In line 329 and elsewhere in the results, include p values/statistics after the word "significant". DONE: In the case of table 3 all parameters were statistically significant, except for the e parameter of the IM/IF cross, so in this case we prefer only to refer to the table 3 for the exact p-values. However, we have put back t and p values for the pairwise comparisons about curve parameters (see Lines 332, 339-341 and 345-348)

L 467-473, Are there SIT programs for pentatomids?  Are you referring generally to the SIT and mass rearing or specifically to cases applied to pentatomids?   Please modify text to improve clarity. DONE: again, very good point (thanks).  By mistake, we included the words “pentatomid” and bugs” in a sentence that should be addressed to the general concept of “Sterile Insect Technique”.  Re-checking everything, we took the opportunity to improve the bibliography too (See Lines 477-479).

This manuscript is a resubmission of an earlier submission. The following is a list of the peer review reports and author responses from that submission.

Round 1

Reviewer 1 Report

The manuscript by Cristofaro and colleagues, reports an investigation on the effects of irradiation on fertility, fecundity and longevity of the Bagrada bug, Bagrada hilaris, an invasive insect pest of economic importance that attack Brassicaceae plant species, corn, kale, arugula, sunflower, among others.  Despite merit and pertinence of the study, I found that the manuscript is misleading from the title, as it gives the erroneous impression that it is about the application of the Sterile Insect Technique (SIT) against B. hilaris when it is not (i.e., while the study tests the effects of irradiation on fecundity, fertility and longevity, this does not represent the application of the SIT).  Authors should note that the SIT involves: (i) mass rearing of the target insect pest, (ii) sterilization (by irradiation) of mass-reared insects, (iii) the release of millions of sterile male insects in the field so that sterile males mate as many times as possible with wild females because females that mate with a sterile male leave no offspring and thus the population of the pest is controlled.  A fundamental component of the SIT is the sexual competitiveness of sterile male insects tested against wild insects, but this was not assessed.  So again, the claim of authors on applying the SIT against B. hilaris is not right.  Authors need to properly flush out the significance of their work.

I have concerns related to the statistical analyses and presentation of results.  It seems to me that the authors unnecessarily complicated the methodological report of the statistics used and that the analyses are not entirely adequate for their type of data. For example, in the cases of data on fertility and fecundity, the best statistical approach is that of an analysis of covariance (which combines elements from a regression and analysis of variance). This is because the experimental design has a continuous predictor variable (irradiation dose from 16 Gy to 120 Gy) and a categorical predictor variable (bug crosses with four levels). The ancova will allow the authors to assess whether: (i) there is a relationship between the irradiation dose and the response variables, (ii) determine whether the different levels of the categorical predictor have different intercepts, and (iii) determine whether the different levels of the categorical predictor have different slopes.  For reference to the authors, please consider the following text which I copy ad verbatim from the fantastic book on statistics by Michael J. Crawley (Crawley. 2007. The R book. John Wiley & Sons. pp. 323-324); I strongly recommend authors to adopt and follow the keys proposed by Crawley for their analyses.  Text taken from Crawley follows (in quotes):

"The hardest part of any statistical work is getting started. And one of the hardest things about getting started is choosing the right kind of statistical analysis. The choice depends on the nature of your data and on the particular question you are trying to answer. The key is to understand what kind of response variable you have, and to know the nature of your explanatory variables. The response variable is the thing you are working on: it is the variable whose variation you are attempting to understand. This is the variable that goes on the y axis of the graph. The explanatory variable goes on the x axis of the graph; you are interested in the extent to which variation in the response variable is associated with variation in the explanatory variable. You also need to consider the way that the variables in your analysis measure what they purport to measure. A continuous measurement is a variable such as height or weight that can take any real numbered value. A categorical variable is a factor with two or more levels: sex is a factor with two levels (male and female), and colour might be a factor with seven levels (red, orange, yellow, green, blue, indigo, violet).

It is essential, therefore, that you can answer the following questions:

• Which of your variables is the response variable?

• Which are the explanatory variables?

• Are the explanatory variables continuous or categorical, or a mixture of both?

• What kind of response variable do you have: is it a continuous measurement, a count, a proportion, a time at death, or a category?

These simple keys will lead you to the appropriate statistical method:

The explanatory variables

(a) All explanatory variables continuous                                               Regression

(b) All explanatory variables categorical                                                Analysis of variance (ANOVA)

(c) Explanatory variables both continuous and categorical            Analysis of covariance (ANCOVA)

The response variable

(a) Continuous                                                                                                 Normal regression, ANOVA or ANCOVA

(b) Proportion                                                                                                  Logistic regression

(c) Count                                                                                                            Log-linear models

(d) Binary                                                                                                           Binary logistic analysis

(e) Time at death                                                                                            Survival analysis"

I recommend authors to indicate clearly and succinctly what are their explanatory and response variables together with their measurement scale in the methods section.  Based on that, on the objectives of the study and on specialized literature on statistics such as Crawley (2007), justify the analyses you used.

Specific comments follow:

In the first paragraph of the introduction, please include a sentence or two indicating what type of dame the bug produces in its host plants.

L 60-61, Suggest deleting "In addition".

L 64-68, More details and background on the SIT are required to better understand the context and the rationale of the study.  Start the paragraph indicating that the SIT is a species-specific and environmentally friendly method of pest control.  Indicate what the technique consists of and how does it work.  Then you can go into the details of irradiation of Hemiptera and application of the SIT against this group of insects.

L 70, Define "sentinel eggs".

L 82-84, Add references to support this statement.

L 88-90, If you mention experiments with other species, please indicate briefly and clearly what those studies conclude.

L 91-94, The authors did not evaluate the suitability of the SIT, they evaluated the effects of irradiation of bugs on their fertility, fecundity and longevity.  Irradiation is a part of the SIT, but it is not right to argue that the suitability of the SIT was evaluated when it was not.

L 96-97, This is irrelevant in the introduction. Move to the acknowledgments section.

L 99-106, Please include more details on the rearing. For example, how many adults started the lab colony, on which host plant(s) they were raised, how does the eggs, nymphs and adults were treated, for how many generations insects were reared in the lab, how many nymphal stages does this species have, etc.

L 118-127, Please explain why fifth instar nymphs were used for irradiation, why not another instar or eggs?  Explain the rationale for using the irradiation doses tested.  Indicate how the nymphs were treated after irradiation; under what conditions nymphs reached the adult stage and sexual maturity?

L 143, give details on the conditions in which eggs hatched.

Figure 1 lacks labels in the y axis.  But as mentioned previously, given the nature of the explanatory variables considered in the experimental design, this figure is not the most appropriate way to present the results.

Figure 3 is more appropriate to present results with the irradiation dose on the x axis as a continuous variable and a fitted lines for each level of the categorical predictor variable.  This figure is nice.  Adopt this style/approach for figure 1 (using ancova as the correct choice of analysis given the nature of predictor variables).

L 311, in the methods section it is indicated that the irradiation was applied to nymphs not "newly emerged adults".

Reviewer 2 Report

Dear authors, the article is interesting, well written and data are clearly presented. However, there are some inaccurancies that requires corrections: why didn't you evaluate longevity of all adults used in the experiments? Or why didn't you evaluate survival of nymphs emerged from all eggs?

Others minor comments and corrections are reported in the attached pdf file.
